# Establishing laboratory-specific reference intervals for TSH and fT4 by use of the indirect Hoffman method

**Sylwia Płaczkowska**[1]*, **Małgorzata Terpińska**[1,2], **Agnieszka Piwowar**[3]

1 Department od Laboratory Diagnostics, Diagnostic Laboratory for Teaching and Research, Faculty of Pharmacy, Wroclaw Medical University, Wroclaw, Poland, 2 Department of Laboratory Diagnostics, University Clinical Hospital, Wroclaw Medical University, Wroclaw, Poland, 3 Department of Toxicology, Faculty of Pharmacy, Wroclaw Medical University, Wroclaw, Poland

* sylwia.placzkowska@umed.wroc.pl

## Abstract

### Background

The results of examinations of laboratory parameters are the basis of appropriate medical decisions. The availability of reliable and accurate reference intervals (RIs) for each laboratory parameter is an integral part of its appropriate interpretation. Each medical laboratory should confirm their RIs. Up-to-date reference intervals for thyroid function hormones are still a matter of ongoing controversy. The aim of the study was the application of the indirect Hoffman method to determine RIs for TSH and fT4 based on the large data pools stored in laboratory information systems and the comparison of these RIs to generally used RIs.

### Material and methods

The TSH and fT4 routine examination results of hospitalized and outpatient populations were collected over five years (2015–2019), and reference limits were established by the improved Hoffmann method after the exclusion of outliers. Comparative verification of established RIs was conducted with the RIs values provided by test manufacturers and literature data.

### Results

Various RIs were observed in different age groups in the examined populations. For TSH, RIs varied between different age groups, with a narrower range of RIs in the studied adult population and a shift of both reference boundaries toward higher values in comparison to manufacturers' data among children. RIs estimated for fT4 were very similar to the manufacturer and literature data.

### Conclusion

Thyroid hormone levels change during a person's lifetime and vary between sexes, but this difference does not always influence the clinical interpretation of laboratory results in the

**Data Availability Statement:** All relevant data are within the manuscript and its Supporting Information files.

**Funding:** The authors received no specific funding for this work.

**Competing interests:** The authors have declared that no competing interests exist.

**Abbreviations:** CDLs, Clinical Decision Limits; CLSI, The Clinical Laboratory Standards Institute; CVa, the intra-series analytical coefficient of variation; CVi, the within-subject biologic coefficient of variation; fT3, free Triiodothyronine; fT4, free Thyroxine; LIS, Laboratory Information Systems; LRL, Lower Reference Limit; RCV, Reference Change Value; RIs, Reference Intervals; TSH, Thyroid Stimulation Hormone; URL, Upper Reference Limit.

context of RIs. The use of indirect methods is justified due to the ease and low cost of their application.

# Background

The results of examinations of laboratory parameters provide useful information for assessing the current health condition of patients. They are necessary for early detection and recognition of disturbances as well as for making appropriate medical decisions. For the purpose of interpretation of laboratory results, the use of the concept of reference intervals (RIs) is currently generally accepted in laboratory medicine [1]. The Clinical Laboratory Standards Institute (CLSI) has released a relevant guideline (C28-A3c) for the evaluation of RIs. According to the CLSI recommendations, an RI is defined as the interval between which 95% of values of a reference population fall into, and includes two extreme reference limits–boundaries derived from the distribution of reference values, which could be associated with good health but also with other physiological or pathological conditions [2, 3]. It is recommended that medical laboratories determine their own local reference intervals to embrace the variations in local populations and the methods and equipment used in particular laboratories. The confusion of RIs with clinical decision limits (CDLs) still remains an issue, especially in paediatric and geriatric age groups, where it presents a significant diagnostic problem [4].

CLSI currently recommends a direct method based on the collection of a minimum of 120 samples from members of a specific preselected reference population, making measurements, and then determining the range which includes 95% of all measured values using a parametric (mean ±2SD) or non-parametric method (2.5th and 97.5th percentile) [2]. The requirement that each laboratory determines its own reference intervals is virtually impossible to perform in practice because of the tremendous amount of time and money required to carry out additional laboratory tests and gather the appropriate reference group. Thus most laboratories adopt external sources for RIs, often without taking into account the problems of transferring values between different populations or laboratory methods [5]. Data provided by the test manufacturer is the most often used source of reference intervals, because such information is required from reagents suppliers by the ISO 15189:2013 standard [6]. At the same time, methods and processes for the determination of reference intervals using indirect methods have been in development for over 50 years, but they are not yet widely applied. This alternative approach is based on the statistical analysis of results generated as part of routine laboratory testing in hospital and outpatient clinics in order to determine reference intervals [3]. Indirect methods eliminate results that do not fit the assumed hypothetical model of the distribution of results–generally a normal distribution–and designate the RI as the central or marginal 95% of the selected results. The application of the indirect method has major potential advantages compared with direct methods. In particular, this process is faster and cheaper; it involves no inconvenience, discomfort, or any additional risk to patients; and laboratory staff need not examine any additional samples. Therefore, additional costs are avoided, which is important in the modern and effective management of the medical laboratory and the hospital [7].

Establishing RIs is particularly problematic for constituents with a large diversity of existing biological variation and inter-population differences, as for example is observed for thyroid hormones, especially thyrotropin–TSH (Thyroid Stimulation Hormone) as well as free triiodothyronine (fT3) and free thyroxine (fT4). The prevalence of thyroid dysfunction in the general world population is estimated to be between 1 and 2% [8]. There are still discrepancies

between TSH, fT3 and fT4 reference values applied to the diagnosis of thyroid dysfunction not only between laboratories, but also in the scientific literature to date. It seems erroneous to apply the concept of universal limits to reference intervals for thyroid function hormones, especially for TSH [9]. Regarding this fact, it appears important and useful to establish reference intervals using a costless, optimized indirect statistical method.

At present, it is assumed that the value of TSH in a healthy general population is approximately 0.4–4.0 mIU/L, which is the result of the fairly high inter-individual variability of this parameter. However, the variability of the value in an individual is much smaller [10, 11], and the value determined in a state of hormonal equilibrium can be regarded as an individual's set-point [12, 13]. Since slight changes in fT4 value correspond to a significant change in TSH, it is used in the screening of disorders of thyroid hormones. Therefore, it is advisable to find out about this individual point by measuring the level of TSH for each person in times of health. This allows for earlier detection of important clinical disturbances in thyroid condition, even without direct comparison to the reference interval [10] or taking into account the physiological changes in concentration related to age [14].

According to the current recommendations, in order to screen for thyroid primary dysfunction, the first TSH determinations should be performed repetitively in 3–6-month intervals, followed by fT4 for differentiation of subclinical and 'overt' thyroid dysfunction. fT3 determinations should be performed only in specific cases [12, 15].

## Material and methods

### Objective

The aim of this study was to establish the reference intervals for TSH and fT4 from the large data pools of patient results stored in laboratory information systems (LIS) using the indirect Hoffman method and to conduct a comparison of RIs with generally used reference limits.

### Laboratory methods

The third generation test of TSH (TSH-3 Ultra) and fT4 (Free Thyroxine) examinations were performed utilising a chemiluminescence method on the Atellica IM analyser (Simens Healthcare Diagnostics Inc., Erlagen, Germany). The linear range of this method was 0.008–150.000 mIU/L and 0.1–12.0 ng/dL for TSH and fT4, respectively. The laboratory intra-series analytical coefficient of variation (CVa) for TSH and fT4 was assessed as 6.4% and 2.5%, and inter-series were 7.1% and 3.2%, respectively.

### Data gathering

The study was performed in accordance with the Declaration of Helsinki and consent was approved by the Wroclaw Medical University Bioethical Commission (decision No. 537/2018). Based on the decision of the bioethics committee, patient informed consent was waived due the retrospective nature of the study conducted on a deidentified aggregated numerical data.

All the laboratory results of TSH and fT4 examinations, together with the patient's age, sex and date of examination, archived in the Laboratory Information System of Department of Laboratory Diagnostics and derived from patients hospitalized at the University Clinical Hospital in Wroclaw during the five-year observation period (1st January 2015– 31st December of 2019 year) were included in the study and used for statistical analysis without any primary selection.

The study included 105 927 TSH results (65 163 from women and 40 764 from men) and 41 400 fT4 results (26 406 from women and 14 994 from men). The participants' age range for the analysed hormones was from 0 to 109 years.

## Statistical analysis

Firstly, before performing any kind of analysis, all data were logarithmically transformed because of the strong right-skewness of the data distribution. Next, all data were divided into 8 age groups ($<1$ y., $\geq 1$ y. $< 6$, $\geq 6$ y., $<12$, $\geq 12$ y. $<18$, $\geq 18$ y. $<40$, $\geq 40$ y. $<65$, $\geq 65$ y. $<90$, $\geq 90$ y.), which reflected the physiological changes associated with human ontogenetic development and the main age groups for which reference values were provided by the manufacturer. The two-sided Tukey test 1.5 IQR was used to reject outliers separately for each age group. The number of excluded records of TSH and fT4 in the studied age groups is reported in detail in Fig 1.

Eventually 100 171 (38 802 men and 61 361 women) and 40 086 (14 508 men and 25 621 women) results for TSH and fT4, respectively, were included in further calculations.

The Hoffman method–an indirect statistical method based on the graphic distribution of lnTSH and lnfT4 values–was applied in each age group for all participants and in regard to sex.

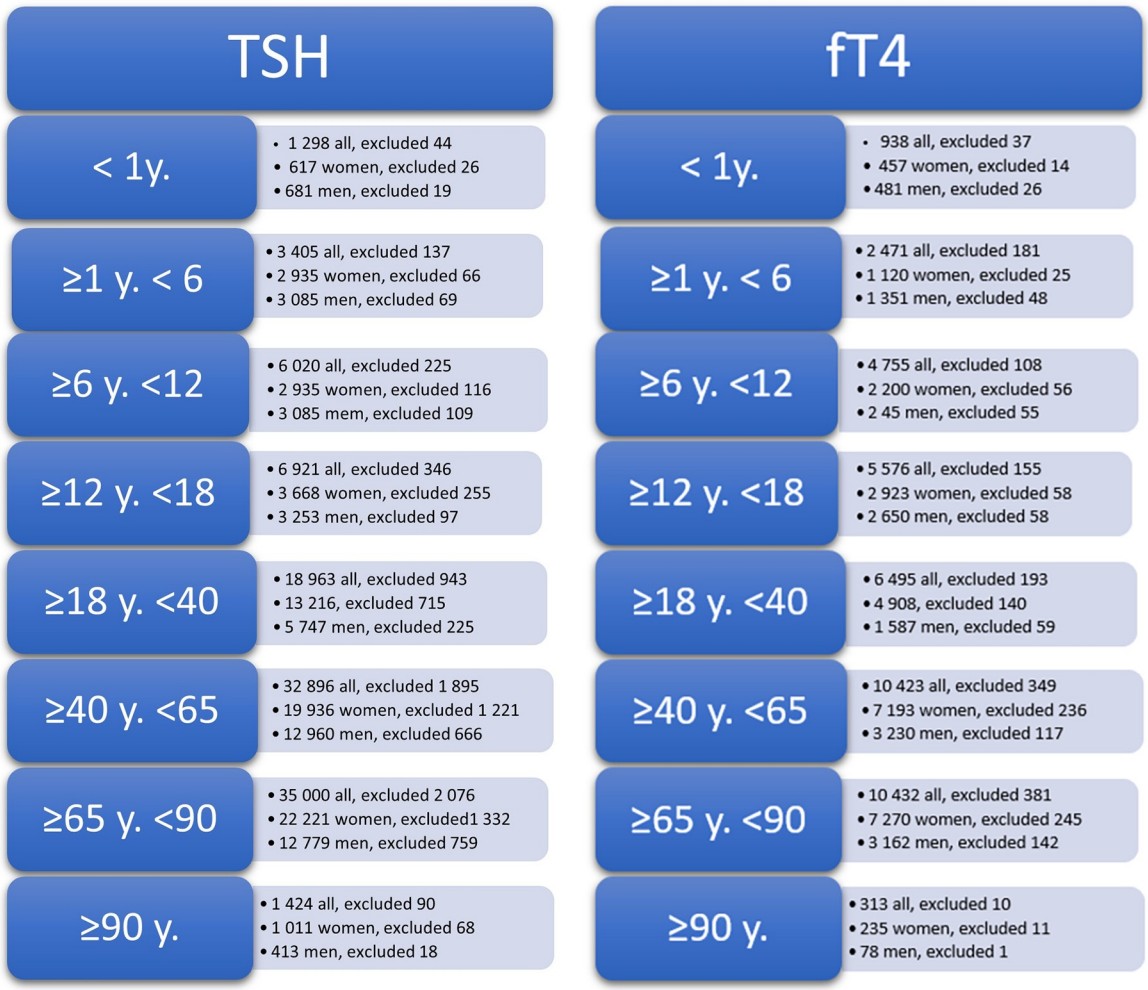

**Fig 1. Number of outliers excluded from the entire data base by the two-sided Tuckey test (1.5IQR) for each studied age group.**

In accord with this method, a Q-Q plot was created in each study age group. The Hoffmann method assumes a Gaussian distribution of physiological test results, and only this range of results is used to determine the reference intervals. On the Q-Q plo, the empirical data with Gaussian distribution creates a straight line with the normal theoretical quantile of the standard normal distribution.

In the next step, an elimination of outliers was visually conducted on the basis of Q-Q plot, and the distribution of remaining data was used to calculate the regression equation using the least squares method. The regression line included the middle range of data, was initially fitted by visual inspection, and was statistically confirmed by determining the linear correlation coefficient. Only r> 0.99 was acceptable. The linear line over the linear part of the Q-Q chart was described by the following equation:

$$y = a \times x + c + e$$

where y = the lnTSH or lnfT4 value, respectively, x = normal theoretical quantile of the standard normal distribution ($\mu = 0$, $\sigma = 1$), a = the slope of the regression line, c = the intercept, e = error.

In the next step, the extrapolation of the linear regression equation to the boundaries of the 95% confidence interval were conducted as follows: *Lower reference Limit* (*LRI*) = $-1.96 \times a + b$ and *Upper reference Limit* (*URI*) = $1.96 \times a + b$. All statistical procedures mentioned above were performed on logarithmically transformed data. The antilogarithm was applied in the last step for the calculation of RIs values on the basis of the linear regression equation.

Linear least squares regression was applied to the middle part of the data distribution. Then Reference Change Value (RCV) was used in order to determine the clinical significance of the relationship between RIs in all selected age groups and manufacturers and published RI. RCV was calculated according to the formula [16]:

$$RCV = 2^{1/2} \times Z \times (CV_a^2 + CV_i^2)^{1/2}$$

where Z is the probability selected for significance, the chosen Z value of 1.96 corresponds to a significance level of 0.05, CVa–the analytic variation (inter-assay variation estimated from our laboratory data) and CVi–the within-subject biologic variation (data from Ricos et al. [17]). In our study, the estimated RCV values for TSF and fT4 were 56.6% and 34.6%, respectively.

Statistical analyses were performed using Statistica 13.1 PL.

## Results

A representative Q-Q plot of the distribution of lnTSH for participants aged $\geq 18$ and $< 40$ is presented in Fig 2. It can be seen that the data included in the analysis for this age group forms a straight line in the middle of the graph and is bent at both ends. This linear range is the basis for the RI estimation for this age group according to the methodology described in the Statistical Analysis section.

The RI values obtained for TSH and fT4 in each separate age group for all study participants and by gender are summarized in Fig 3.

For TSH, the estimated values of RIs decreased with the age of the patients and simultaneously tended to decrease ranges. An inverse relationship was observed for the fT4 value with regard to the RI range, which increased with age, while the RI limits did not indicate significant shifts in values. Detailed TSH numerical data related to Fig 3 for all results are provided together with a comparison of LRI and URI for women and men in Table 1.

The applied Hoffman method shows various TSH RIs in different age groups. The LRIs gradually decreased in subsequent age groups, from children up to the age of $\geq 90$; for URI, the

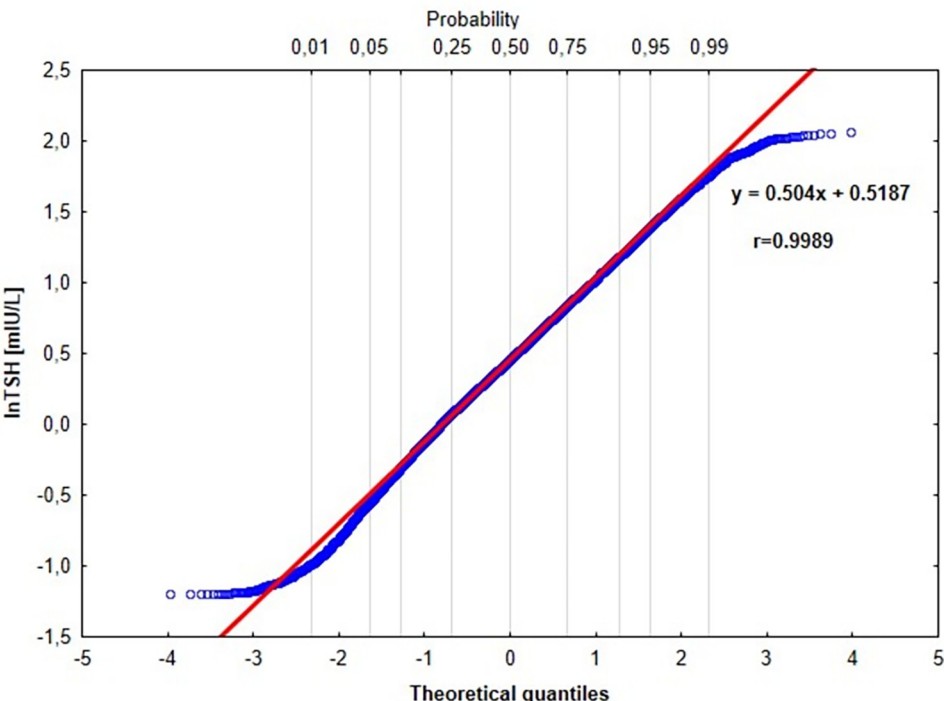

**Fig 2. A representative Q-Q plot of distribution of lnTSH for participants aged ≥18 and <40.**

same trend was observed up to <65 and then increased in participants older than 65. Differences greater than those determined by RCV for comparisons between each age group were revealed for LRI in all subjects as well as in separate samples of women and men. In all cases, they concerned the results of the oldest patients, children and adolescents.

The comparison of LRI and URI between women and men in particular age groups showed the greatest difference for LRI of 11.7% TSH in the age group of <1 years. The same analysis of URI for TSH indicated the highest differences in group > 90 years, but they were also among the highest among infants. It is worth mentioning that the obtained differences in percentage for LRI and URI are smaller than the extra-individual variability for TSH– 24.9% [16]. Therefore, the results obtained for all study participants in a given age group were considered as general RI values, and differences in sex were not taken into account.

The LRIs for participants aged <40 established in this study were within TSH RIs provided by the manufacturer. For the three oldest participant groups, the LRIs were lower than reported by manufacturers, while TSH were higher in children and adolescents and groups ≥65 years. For adults, URI estimated by Q-Q plots were lower than provided by the manufacturer. Generally, in comparison to the manufacturers' data, the Hoffman method revealed a shift of both reference boundaries toward higher in children and adolescents, narrower RIs for TSH in the studied adult population, and wider RIs boundaries among seniors. The difference between LRI and URI established by the Hoffman method with RIs reported by the manufacturer did not exceed the acceptable 56% RCV for TSH in any age group (Table 2).

RIs established for fT4 presented in Table 3 were characterized by low variability between different sex and age groups.

The sex differences for fT4 LRI were less than 10%, and the highest were observed in infants and age groups ≥40 y. <65 and ≥ 90 y. In the youngest group, the difference in URI between

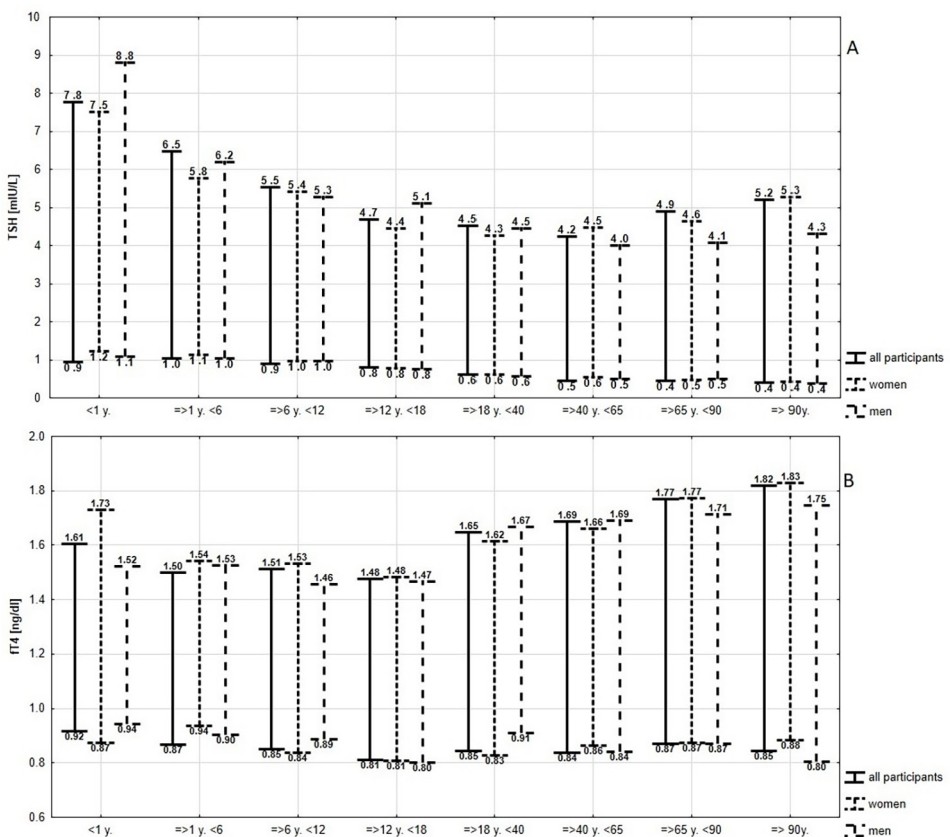

**Fig 3.** RIs for TSH (panel A) and fT4 (panel B) in all participants included in the analysis groups and among males and females in each age group.

sexes was lower than even 5%, which is only a quarter of the RCV value established for fT4 in this study. Therefore, further analyses were carried out without regarding sex.

The differences calculated for comparison between subsequent age groups were below the value of RCV 21.6% in all cases. It is worth to mentioning the obtained differences in percentage for LRI and URI are smaller than the extra-individual variability of fT4–12.1% [16].

**Table 1. Estimation of reference intervals with the Q-Q plot method for all TSH results and percentage differences in TSH reference between women and men.**

| age group | N records included to the final analysis | all participants | | LRI | | | URI | | |
|---|---|---|---|---|---|---|---|---|---|
| | | LRI | URI | women | men | % of difference | women | men | % of difference |
| < 1 y. | 1254 | 0.93 * | 7.76 | 1.23 * | 1.08 * | 11.7 | 7.51 | 8.81 | -17.3 |
| ≥1 y. < 6 | 3268 | 1.03 † | 6.48 | 1.13 † | 1.04 † | 7.3 | 5.77 | 6.18 | -7.2 |
| ≥6 y. <12 | 5795 | 0.91 ‡ | 5.54 | 0.96 | 0.97 ‡ | -0.5 | 5.42 | 5.28 | 2.6 |
| ≥12 y. <18 | 6575 | 0.79 | 4.68 | 0.77 | 0.76 | 2.1 | 4.44 | 5.11 | -15.0 |
| ≥18 y. <40 | 18020 | 0.63 | 4.51 | 0.62 | 0.56 | 9.2 | 4.25 | 4.46 | -4.8 |
| ≥40 y. <65 | 31001 | 0.45 | 4.23 | 0.55 | 0.49 | 10.4 | 4.48 | 4.01 | 10.6 |
| ≥65 y. <90 | 32924 | 0.44 b | 4.89 | 0.48 *,† | 0.49 | -2.9 | 4.64 | 4.07 | 12.2 |
| ≥90 y. | 1334 | 0.39 *,†,‡ | 5.20 | 0.43 *,† | 0.39 *,†,‡ | 8.9 | 5.26 | 4.31 | 18.1 |

*, †, ‡- RCV between age group *: <1 year of age, †: ≥1 year of age <6, ‡: ≥years of age <12, and another age group marked with the same symbol within the analyzed group of participants (columns) exceeds 56.0%.

**Table 2. Comparison of calculated TSH reference intervals with RIs reported by manufacturer.**

| TSH [mIU/L] Hoffman method | | | | Manufacturer RI 2.5–97.5 percentile | | | | difference (%) between established and manufacturer RI* | |
|---|---|---|---|---|---|---|---|---|---|
| Age group | N | LRI | URI | Age group | N | LRI | URI | LRI | URI |
| < 1 y. | 1254 | 0.93 | 7.76 | <2 y. | 94 | 0.87 | 6.15 | 6.4 | 20.7 |
| ≥1 y. < 6 | 3268 | 1.03 | 6.48 | 2–12 y. | 198 | 0.67 | 4.16 | 34.9 | 35.8 |
| ≥6 y. <12 | 5795 | 0.91 | 5.54 | | | | | 26.4 | 24.9 |
| ≥12 y. <18 | 6575 | 0.79 | 4.68 | 13–20 y. | 150 | 0.48 | 4.17 | 39.2 | 10.9 |
| ≥18 y. <40 | 18020 | 0.63 | 4.51 | adults | 229 | 0.55 | 4.78 | 12.7 | -6.0 |
| ≥40 y. <65 | 31001 | 0.45 | 4.23 | | | | | -22.2 | -13.0 |
| ≥65 y. <90 | 32924 | 0.44 | 4.89 | | | | | -25.0 | 2.2 |
| ≥90 y. | 1334 | 0.39 | 5.20 | | | | | -41.0 | 8.1 |

* accepted TSH RCV <56%.

A comparison of calculated fT4 reference intervals with RIs reported by the manufacturer is presented in Table 4.

LRI for fT4 were almost the same as the ranges provided by manufacturers, and higher differences were noticed for infants' URIs. Generally, LRIs were identical for infants and children, while the URI was slightly higher than provided by manufacturer. Narrower RIs falling within the manufacturers' data were observed for the adolescent group. A minimal shift toward the lower values of RIs were revealed for adults aged ≥40 and < 65 years. The similar trend was observed for seniors, but only for the LRI value.

## Discussion

The conducted analysis of TSH and fT4 RIs using the Hoffman method revealed that the differences between the sexes in LRI and URI values in individual age groups are smaller than the assumed RCV value. When comparing the age groups as a whole and taking sex into account, RI showed differences exceeding the accepted critical value only for TSH between the youngest and the oldest participants in the study.

Currently, medical decisions are mainly based on results of laboratory diagnostic tests, which are used to confirm, exclude, classify or monitor disease in order to guide treatment. Therefore, establishing appropriate reference ranges is crucial for the correct interpretation of

**Table 3. Estimation of reference intervals with the Q-Q plot method for all fT4 results and percentage differences in TSH reference between women and men.**

| age group | N records included to the final analysis | all participants | | LRI | | | URI | | |
|---|---|---|---|---|---|---|---|---|---|
| | | LRI | URI | women | men | % of difference | women | men | % of difference |
| < 1 y. | 901 | 0.92 | 1.61 | 0.87 | 0.94 | -7.9 | 1.73 | 1.52 | 12.1 |
| ≥1 y. < 6 | 2390 | 0.87 | 1.50 | 0.94 | 0.90 | 3.5 | 1.54 | 1.53 | 1.0 |
| ≥6 y. <12 | 4647 | 0.85 | 1.51 | 0.84 | 0.89 | -5.7 | 1.53 | 1.46 | 5.0 |
| ≥12 y. <18 | 5418 | 0.81 | 1.48 | 0.81 | 0.80 | 1.0 | 1.48 | 1.47 | 1.1 |
| ≥18 y. <40 | 6302 | 0.85 | 1.65 | 0.83 | 0.91 | -9.8 | 1.62 | 1.67 | -3.1 |
| ≥40 y. <65 | 10074 | 0.84 | 1.69 | 0.86 | 0.84 | 2.5 | 1.66 | 1.69 | -1.7 |
| ≥65 y. <90 | 10051 | 0.87 | 1.77 | 0.87 | 0.87 | 0.2 | 1.77 | 1.71 | 3.5 |
| ≥90 y. | 303 | 0.85 | 1.82 | 0.88 | 0.80 | 8.8 | 1.83 | 1.75 | 4.6 |

**Table 4. Comparison of calculated fT4 reference intervals with RIs reported by manufacturer.**

| fT4 [ng/dl] Hoffman method | | | | Manufacturer RI 2.5–97.5 percentile | | | | absolute difference (%) between established and manufacturer RI* | |
|---|---|---|---|---|---|---|---|---|---|
| Age group | N | LRI | URI | Age group | N | LRI | URI | LRI | URI |
| <1 y. | 901 | 0.92 | 1.61 | <2 y. | 72 | 0.94 | 1.44 | 2.2 | 10.5 |
| ≥1 y. < 6 | 2 390 | 0.87 | 1.50 | 2–12 y. | 190 | 0.86 | 1.40 | 1.1 | 6.7 |
| ≥6 y. <12 | 4 647 | 0.85 | 1.51 | | | | | -1.1 | 7.3 |
| ≥12 y. <18 | 5 418 | 0.892 | 1.326 | 13–20 y. | 129 | 0.83 | 1.43 | 6.9 | 7.8 |
| ≥18 y. <40 | 6 302 | 0.85 | 1.65 | adults | 388 | 0.89 | 1.76 | -4.7 | -6.7 |
| ≥40 y. <65 | 10 074 | 0.84 | 1.69 | | | | | -5.9 | -4.1 |
| ≥65 y. <90 | 10 051 | 0.87 | 1.77 | | | | | -2.3 | 0.6 |
| ≥90 y. | 303 | 0.85 | 1.82 | | | | | -4.7 | -3.2 |

* accepted fT4 RCV < 21,6%.

laboratory results. Unfortunately, our observations show that the main source of reference intervals in Poland is data from literature, often based on research carried out among the general population or with a different genetic and/or cultural profile [18]. The second source of information about the expected values is provided by the laboratory reagent manufacturer. The analysis of the RI values in these materials shows that although the manufacturers declare that their procedures comply with the CLSI recommendations, the reference groups are very often too small in number and very poorly characterized in many important aspects, such as race, age group, health status, or body weight [19–21]. Very rarely, a laboratory procedure is also used in order to verify the reference intervals provided by the manufacturer according to CLSI recommendations [2]. The RIs for TSH determined in our study by Hoffman method show a significant agreement with RI in the age group ≥18 y. <40 and URI in adolescents and adults provided by the manufacturer. In other age groups, the RI differences, although they did not exceed the RCV, were higher than 20%. However, with regard to fT4, the observed differences in the values determined by the Hoffman method and provided by the manufacturer were minimal and did not relate to any specific age groups. This indicates the possibility of using the RIs provided by the manufacturer, but this would require initial confirmation. Confirmation of the manufacturer's RIs compliance by CLSI recommendations requires the completion of additional laboratory tests in a strictly defined reference group of at least 20 people for each age range and, if required, taking sex into account. A comparison of RIs established by an individual laboratory with the manufacturer's value excludes the differentiation of the test results caused by a different measurement system, but there are still differences resulting from pre-analytical conditions and population differences. The use of the Hoffman method as a method of determining reference intervals does not require additional tests, and at the same time allows for the determination of any age intervals, not only those proposed by the manufacturer.

Determination of TSH is currently the main parameter used in the screening of thyroid disorders and as a therapeutic target and prognostic marker. This approach requires the establishment of specific reference limits, not only for TSH but also for other thyroid hormones, which are in close physiological relationship [13]. Numerous studies have been conducted worldwide to set such limits for the general population, but there is still no consensus on this matter [22, 23]. This is due to numerous preanalytical and analytical factors of thyroid hormone

**Table 5. Comparison of reference intervals for TSH and fT4 obtained by different indirect methods.**

| Author, year | method of RI reported by author | age group (n) | LRI—URI |
|---|---|---|---|
| **TSH [units]** | | | **[mIU/L]** |
| Mokhatar KM [24]; 2020 | Batattacharya method | >18 years (8838) | 0.44–4.4 |
| | Quantile regression with RCQ | 18–29 (1618) | 0.46–3.9 |
| | | 30–39 (1981) | 0.44–4.1 |
| | | 40–49 (1769) | 0.42–4.4 |
| | | 50–59 (1670) | 0.41–4.5 |
| | | 60–69 (1320) | 0.39–4.5 |
| | | >70 (480) | 0.38–4.2 |
| Lo Sasso et al. [5]; 2019 | Reference Limit Estimator Software | 15–105 years | 0.18–3.54 |
| | | (22,602) | |
| | | women | 0.18–3.94 |
| | | (12,099) | |
| | | men | 0.19–3.23 |
| | | (7805) | |
| Drees et al. [25]; 20 | software EP Evaluator (Data Innovations) | 0.5–2 (417) | 0.60–5.28 |
| | | 2–10 (3377) | 0.72–4.92 |
| | | 11–17 (8001) | 0.55–4.42 |
| | | 18–49 (7563) | 0.50–4.00 |
| | | 50–64 (6511) | 0.48–4.37 |
| | | 65–79 (5314) | 0.54–4.84 |
| | | >80 (1784) | 0.54–5.31 |
| Stich et al. [26];2015 | Hoffman followed by log transformation | 11–14 (1377) | 1.43–4.21 |
| | | 19–30 (4416) | 1.08–4.40 |
| Larisch et al. [27];2015 | improved Hoffmann and Katayev's method | adult subjects (399) | 0.57–3.32 |
| Feng et al. [28]; 2014; | improved Hoffmann and Katayev's method | 25–85 years (10870) | 0.233–4.979 |
| Dorizzi et al. [29];2011 | improved Hoffmann and Katayev's method | >18 years (21,862) | 0.16–3.28 |
| Katayev et al. [30]; 2010 | improved Hoffmann and Katayev's method | >18 years (129,443) | 0.45–3.05 |
| **fT4 [units]** | | | **[ng/ml]** |
| Dittadi et al. [31]; 2021 | Batattacharya method | - | 0.61–1.14 (7.93–14.69 pmol/L) |
| Kapelari et al. [32]; 2008 | posteriori direct methods 2.5th to 97.5th | infants (45) | 0.71–1.97 (9.17–25.28 pmol/L) |
| | | 6–10 years (327) | 0.82–1.62 (10.60–20.90 pmol/L) |
| | | 15–18 years (233) | 0.82–1.78 10.57–22.62 (pmol/L) |

determinations and population differences [5, 9]. There are a limited number of publications available using the indirect method (e.g. Hoffman method) for RI estimation in the general population, and the results of most of them concerning TSH and fT4 are summarized in Table 5.

The list of values obtained by indirect methods presented in Table 5 shows a significant differentiation in both LRI and URI and the width of the reference interval for TSH. The LRI obtained by the modified method of Hoffman and Katayev ranges from 0.16 to 1.08 mIU/L, and the URI values range from 3.05 to 4.98 for adults. The results obtained in our study are the closest to the results obtained by Drees et al. [25] by software EP Evaluator (Data Innovations) and Mokhatar KM [24] by Quantile regression with RCQ method. Hoffman's method was not used in any of the available studies to determine the reference intervals for fT4. However, the available results of two studies using other indirect methods [31, 32] show a fairly good agreement between each other and the results of our research.

The determination of reference intervals for thyroid hormones is the subject of ongoing debate due to poor standardisation of immunochemical methods and the lack of unambiguous

reference materials [33] as well as large inter-population differences, along with variability related to the age and sex of patients within the same population. The legitimacy of determining reference intervals for thyroid hormones with an indirect method based on the results of hospital tests is additionally supported by the fact that they are very often acquired from screening tests, and therefore a large portion of the results comes from people without thyroid disorders. The use of such an approach additionally eliminates the differences in values that may result from different methods of sampling in different hospital wards and collection points [25].

Encouraged by the hints contained in Jones' publication [3], in the field of disseminating the results of research on determining reference intervals, regardless of the applied approach: direct or indirect, we searched for the values of reference intervals characteristic for the population of our hospital and the methods and apparatus used in our laboratory. At the same time, we are aware that no intervals are perfect and final, and the results obtained by the indirect method, even if they are not absolutely accurate, are closer to the actual state of the population of a given region, because they take into account the analytical and biological variability of the analysed parameter.

## Conclusions

TSH and fT4 levels change during a person's lifetime and vary between sexes, but the difference does not always have an influence on the clinical interpretation of laboratory results in the context of reference intervals. The discussed differences between RIs estimated in this study and the literature data as well as manufacturers' information reflect the actual distribution of results in the population of a given region and are consistent with the idea of screening the functioning of the thyroid gland firstly by TSH and fT4. Differences between our RIs in comparison to other direct and indirect studies were probably caused by analytical (different antibody characteristics used in reagents, lack of harmonization) and epidemiological factors (different populations, socio-economic status and undefined geographic covariates).

### Strengths of the study

Considering common weaknesses or even shortcomings in determining reference intervals, such as the uncritical acceptance of the values supplied by the manufacturer, indirect methods with their implementation of computer applications [18] may well be the best alternative for regional hospitals and field laboratories serving the population of a given region. This is especially useful in terms of accessing hard-to-reach paediatric or geriatric populations.

Comparing the RIs obtained by the indirect method can also be used to confirm compliance with the RI values provided by the manufacturer of the reagent kits. Considering the differentiation of reference values determined both by direct and indirect methods in different populations, the use of indirect methods is, in our opinion, justified due to their low cost, ease of application, and above all the possibility of imaging the distribution of results in the population of a given region rather than relying on data from other geographic and cultural areas.

### Limitations of the study

The greatest limitation of this study was using unselected data from our LIS. We did not have the possibility to download the data with the ICD code and delete results repeatedly performed on the same patient. However, the applied Hoffman method is based on the use of values located in the middle of the distribution to estimate the reference intervals. This was preceded by the rejection of outliers, i.e. pathological values–the Hoffman method is relatively robust in their occurrence. Another important limitation was a lack of verification of the indirect RIs

established using the direct method which would have enabled us to confirm that the exclusion of patients with thyroid disorders is not required to obtain proper reference intervals from hospital populations with many thousands of records.

## Supporting information

**S1 Dataset.**
(XLSX)

## Author Contributions

**Conceptualization:** Sylwia Płaczkowska, Agnieszka Piwowar.

**Data curation:** Sylwia Płaczkowska, Małgorzata Terpińska.

**Formal analysis:** Sylwia Płaczkowska.

**Investigation:** Małgorzata Terpińska, Agnieszka Piwowar.

**Methodology:** Sylwia Płaczkowska.

**Resources:** Małgorzata Terpińska.

**Writing – original draft:** Sylwia Płaczkowska.

**Writing – review & editing:** Sylwia Płaczkowska, Małgorzata Terpińska, Agnieszka Piwowar.

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
