## [Decision Letter · Decision Letter 0]

8 Oct 2021

PONE-D-21-26253Establishing laboratory-specific reference intervals for TSH and fT4 by use of the indirect Hoffman method PLOS ONE

Dear Dr. Płaczkowska,

Thank you for submitting your manuscript to PLOS ONE. After careful consideration, we feel that it has merit but does not fully meet PLOS ONE’s publication criteria as it currently stands. Therefore, we invite you to submit a revised version of the manuscript that addresses the points raised during the review process.

Two referees and I have reviewed your manuscript (MS) and both referees have requested changes and additions for you to address in a revised MS.  I find all comments pertinent, from both referees, especially those by Ref. 1 about providing a more complete methods section and a more readable Discussion, and reorganizing some topics.  Ref. 1 has assessed your statistical analysis and found it adequate, so the comment about that by Ref. 2 can be ignored.  Your attention to major and minor comments should make your contribution more correct and impactful.

We look forward to receiving your revised manuscript.

Kind regards,

Joseph DiStefano III, PhD

Academic Editor

PLOS ONE

Journal Requirements:

Reviewers' comments:

Reviewer's Responses to Questions

**Comments to the Author**

1. Is the manuscript technically sound, and do the data support the conclusions?

Reviewer #1: Yes

Reviewer #2: Yes

2. Has the statistical analysis been performed appropriately and rigorously? 

Reviewer #1: Yes

Reviewer #2: I Don't Know

3. Have the authors made all data underlying the findings in their manuscript fully available?

Reviewer #1: No

Reviewer #2: No

4. Is the manuscript presented in an intelligible fashion and written in standard English?

Reviewer #1: Yes

Reviewer #2: Yes

5. Review Comments to the Author

Reviewer #1: General Comment

The authors used an indirect method to establish reference intervals for TSH and FT4. This is an interesting approach, which merits greater attention in the literature. The method offers great potential to clinicians to efficiently validate reference intervals provided by manufacturers and laboratories using data readily available to them from a local population of interest.

As for the strength of Methods, the description of methods is incomplete, lacking details which are necessary for the reader to follow the process from data collection to final results. Some of the critical omissions are discussed but belong to methods in the first place. The structure of the paper can be improved to better guide the reader through the whole article. Suggest to make some statements more precise, elaborate on methodological detail and summarise results in the text. The discussion could be more concise and better structured. Specific comments are provided below.

Introduction

Line 104 TSH This is not generally true, approximately 0.4 to 4 might be the best we can rightfully claim to be precise.

Line 105 "variability of the value in an individual" Pls provide a reference for such statements, e.g. Andersen et al., also pls for "setpoint" in the next line.

Line 110 "earlier detection of disturbances". This is certainly true, but when would these disturbances indicate true change in the underlying thyroid condition to the clinician rather than fluctuations.

Line 112 "in order to confirm thyroid primary dysfunction" This is wrong, and in need of a reference. For confirmation, unlike screening, all three thyroid hormones may be required, because there is diseases such as T3 hyperthyroidism.

Line 115 "it is not advisable to order isolated fT4 testing" This is a misrepresentation of the reference. Again, true for screening only. Otherwise, FT4 may be useful by itself in secondary hypothyroidism where TSH may fail to confirm the diagnosis.

These statements do not provide compelling reasons, but it is ok for the authors to focus on the two main hormones if they wish to do so.

Line 121 "to provide the possibility to utilise" This is difficult to understand. Pls simplify, perhaps "to evaluate"

Why use the method of Hoffman? Why not prefer Katayev?

Methods

Line 128 Atellica IM The manufacturer of the assay should be named.

What was the inter-series performance of the methods?

What was the conventional reference range for TSH and FT4?

Line 137 "completely anonymous" This is not exactly right terminology as per data protection standards, probably the authors meant "deidentified aggregated data".

Line 143 "Excel spreadsheet" Excel is not regarded as a proper data base solution, and not considered safe (even banned) by some institutions and organisations.

Line 149 Has the diagnostic code (ICD) been considered in selecting patients? What happened to severely ill patients or patients with interfering comorbidities and medications? What about the use of thyroid medication? What happened to repeat measurements from the same patient? These are critical issues that belong to methods.

If the authors felt their method is more robust than other procedures, for instance requiring less accurate sampling of normal subjects, this would be most interesting, but should be included in the evaluation of the method.

Statistical analysis

It has been reported that log transformation failed to achieve both linearity of TSH and an acceptable normal distribution unless TPO-Ab positive subjects who add a right skew are removed prior to the analysis.

Did the authors examine a possible influence by contamination of the euthyroid sample, for instance with auto-antibody positive subjects?

Clinical categorisation is essential when establishing a reference range, which does not conceptionally extend to heterogeneous populations with various thyroid pathologies.

The appropriateness of the age groups should be evaluated with age as a continuous outcome to further confirm their physiologically-based selection by cut-off analysis. Given the large sample size this should be feasible.

Tuckey test? Pls specify at what levels patients were excluded.

The Hoffman method needs to be explained more in detail and referenced.

Why Hoffman? What is the difference to the method of Katayev el al.. The latter provides a detailed statistical procedure for dealing with the error term.

FT4 does follow a normal distribution. Hence, a log transformation seems statistically unnecessary, and clinically unwanted, even potentially detrimental given the well known issues with interpretation of back transformed estimates in clinical medicine.

Figure 1 should be moved to the Results section. The extrapolation to the boundaries should be precisely indicated in the Figure, as this is a critical part of the method.

A statistical measure of how close the data are to the fitted line such as r squared should be reported.

Results

Line 192 A brief summary of main outcomes beyond merely referring to tables would be more informative to the reader, e.g. comment briefly on age dependency.

Line 196 "decreased" with age" This could be more informative, for instance say something about the magnitude or relevance of effects. In a large sample there will always be some minor differences, which might not be all that relevant.

Line 200 "significant" Does this mean statistically significant?

Have differences been statistically assessed, for instance by interaction with age or age categories?

Discussion

The authors briefly discuss the method by Katayev et al.. This method has been successfully adopted by a few authors in the literature to verify TSH reference values, compare them with the limits provided by manufacturers and, importantly, facilitate clinical decision making in borderline conditions. The authors might want to consider discussing/ replicating some relevant findings from those publications, some of which they did not mention.

The discussion is detailed, but difficult to follow. For instance in line 346 the authors mention Ris and then a comparison with LRI.

A clearer structure and concise summary of similarities/ differences to other approaches would help.

Line 403 "therefore a large portion of the result comes from people without thyroid disorders." This raises some questions to be both examined (see Methods) and discussed. What impact do thyroid disorders have on the validity of the method? At what proportion of pathologies might the method fail? What makes the method robust?

Figures

I can’t seem to find legends to Figures.

Reviewer #2: The authors have defined reference intervals (RIs) for TSH and FT4 by applying the indirect Hoffman method based on a large data pools from patients and outpatients stored in laboratory information systems over a period of five years. There are not many data regarding the Hoffman method and thus, they can be compared only with manufacturers’ data. It is an analytical study. The methods should be reviewed by a statistician who has experience with Hoffman’s method.

The study is interesting and methodologically novel. However, I have some minor comments:

1. Line 400: “The legitimacy of determining reference intervals for thyroid hormones with

an indirect method…based on the results of hospital tests is additionally supported by

the fact that these tests are screened, and therefore a large portion of the results comes from people without thyroid disorders”. However, this method does not eliminate differences in the levels of parameters (TSH & FT4) resulting from drugs and concomitant diseases which are common in hospital sampling and interfere with TSH and particularly with FT4 measurement?

2. The data should be validated by other studies and compared with direct methods.

The URIs and LRIs estimated by the indirect method may be compared for concordance with the 2.5th and 97.5th percentile analysis? This evaluation can be performed easily and cheaply as validation of Hoffman’s method.

3. The indirect method however, may not distinguish well enough pathological from non-

pathological levels and this may be an additional limitation.

4. This analysis clearly shows the importance of measuring FT4 together with TSH in

establishing a diagnosis of thyroid disease. TSH is a reliable indicator of disease.

5. A minor linguistic revision is needed.

6. PLOS authors have the option to publish the peer review history of their article (what does this mean?). If published, this will include your full peer review and any attached files.

Reviewer #1: **Yes: **Rudolf Hoermann

Reviewer #2: **Yes: **Leonidas Duntas

---

## [Author Response · Author response to Decision Letter 0]

15 Nov 2021

Dear Reviewer’s 

Thank you very much for all your constructive comments concerning our manuscript. All changes are highlighted in yellow in revised version. A detailed description of changes in the manuscript appears below.

Reviewer #1: General Comment

We kindly thank you for giving us the chance to improve our manuscript. Thank you very much for all your constructive comments concerning our manuscript. A detailed description of changes in the manuscript appears below.

The authors used an indirect method to establish reference intervals for TSH and FT4. This is an interesting approach, which merits greater attention in the literature. The method offers great potential to clinicians to efficiently validate reference intervals provided by manufacturers and laboratories using data readily available to them from a local population of interest.

As for the strength of Methods, the description of methods is incomplete, lacking details which are necessary for the reader to follow the process from data collection to final results. Some of the critical omissions are discussed but belong to methods in the first place. The structure of the paper can be improved to better guide the reader through the whole article. Suggest to make some statements more precise, elaborate on methodological detail and summarise results in the text. The discussion could be more concise and better structured. Specific comments are provided below.

Introduction

Line 104 TSH This is not generally true, approximately 0.4 to 4 might be the best we can rightfully claim to be precise.

Answer:

Thank you for drawing our attention to this statement. We decreased the firmness of the quoted values in relation to the general population by adding the word “approximately”, bearing in mind the discrepancies in the literature which are generally summed up as the range 0.4-4.0 and now it sounds “is approximately 0.4-4.0 mIU/L”

Line 105 "variability of the value in an individual" Pls provide a reference for such statements, e.g. Andersen et al., also pls for "setpoint" in the next line.

Answer:

As suggested by the reviewer, we introduced appropriate literature references in the places indicated by the reviewer:

- Andersen S, Pedersen KM, Bruun NH, Laurberg P. Narrow individual variations in serum T(4) and T(3) in normal subjects: a clue to the understanding of subclinical thyroid disease. J Clin Endocrinol Metab. 2002 Mar;87(3):1068-72. doi: 10.1210/jcem.87.3.8165. PMID: 11889165

-Hoermann R, Midgley JEM, Larisch R, Dietrich JW. Recent advances in thyroid hormone regulation: toward a new paradigm for optimal diagnosis and treatment. Front Endocrinol (Lausanne). 2017;8:364. https://doi.org/10.3389/fendo.2017.00364

Now this passage is as follows: “However, the variability of the value in an individual is much smaller [10] , and the value determined in a state of hormonal equilibrium can be regarded as an individual's set-point [11].”

Line 110 "earlier detection of disturbances". This is certainly true, but when would these disturbances indicate true change in the underlying thyroid condition to the clinician rather than fluctuations.

Answer:

We have edited the sentence indicated by the reviewer and we hope that it is currently correct. The sentence reads as follows:

“This allows for earlier detection of clinical important disturbances in thyroid condition, even without direct comparison to the reference interval and taking into account the physiological changes in concentration related to age.”

Line 112 "in order to confirm thyroid primary dysfunction" This is wrong, and in need of a reference. For confirmation, unlike screening, all three thyroid hormones may be required, because there is diseases such as T3 hyperthyroidism.

Answer:

We have edited this paragraph in line with the reviewer's instructions. Now this passage is as follows: “This allows for earlier detection of clinical important disturbances in thyroid condition, even without direct comparison to the reference interval [12] and taking into account the physiological changes in concentration related to age [13].”

Line 115 "it is not advisable to order isolated fT4 testing" This is a misrepresentation of the reference. Again, true for screening only. Otherwise, FT4 may be useful by itself in secondary hypothyroidism where TSH may fail to confirm the diagnosis.

These statements do not provide compelling reasons, but it is ok for the authors to focus on the two main hormones if they wish to do so.

Answer:

We have edited this paragraph in line with the reviewer's instructions and now it reads:

“According to the current recommendations, in order to screening for thyroid primary dysfunction, first TSH determinations should be performed repetitively in 3-6 month intervals, followed by fT4 for differentiation of subclinical and “overt” thyroid dysfunction. fT3 determinations should be ordered only in specific cases [12-15].”

Line 121 "to provide the possibility to utilise" This is difficult to understand. Pls simplify, perhaps "to evaluate"

Answer:

We have edited the Objective paragraph in line with the reviewer's instructions and now it reads:

“The aim of this study was to establishing the reference intervals for TSH and fT4 from the large data pools of patient results stored in laboratory information systems (LIS) using the indirect Hoffman method, and conducting a comparison of RIs to generally used reference limits.”

Why use the method of Hoffman? Why not prefer Katayev?

Answer:

We decided to use the Hoffman method with insignificant modifications consisting in logarithmic data transformation due to its simplicity and the possibility of using simple spreadsheets, e.g. Excel (Georg Hoffmann et al. in 2016 (DOI 10.1515/labmed-2015-0104) or statistical packages such as Statistica in our case. Katayeva's computerized method is definitely more complicated methodically and requires the use of the R package, which is actually free but definitely more complicated to use, especially for employees of routine medical laboratories. More detailed explanations of the choice of method are provided in response to Statistical Analysis.

Methods

Line 128 Atellica IM The manufacturer of the assay should be named.

What was the inter-series performance of the methods?

What was the conventional reference range for TSH and FT4?

Answer:

We have completed the missing information on the reagent supplier and the inter-series variability. Whereas, the conventional reference ranges used in our laboratory were the same as the manufacturer provided and showed at Tables 2 (TSH) and Table 4 (fT4).

Line 137 "completely anonymous" This is not exactly right terminology as per data protection standards, probably the authors meant "deidentified aggregated data".

Answer:

We entered the correct nomenclature and now the sentence is:

“Based on the bioethics committee decision patient informed consent was waived due the retrospective nature of the study conducted on a deidentified aggregated numerical data.”

Line 143 "Excel spreadsheet" Excel is not regarded as a proper data base solution, and not considered safe (even banned) by some institutions and organisations.

Answer:

In our institution, it is acceptable to use Excel as a database with safety standards appropriate for our conditions. However, due to the doubts that arose, we have removed this passage from the manuscript.

Line 149 Has the diagnostic code (ICD) been considered in selecting patients? What happened to severely ill patients or patients with interfering comorbidities and medications? What about the use of thyroid medication? What happened to repeat measurements from the same patient? These are critical issues that belong to methods.

If the authors felt their method is more robust than other procedures, for instance requiring less accurate sampling of normal subjects, this would be most interesting, but should be included in the evaluation of the method.

Answer:

We are aware that no method of determining RI is perfect and it is always a choice between a well-defined but not very numerous group in direct methods and a less characterized but definitely more numerous group in indirect methods. In our opinion, any effort made to best match the reference interval to the cared for patient population is better than adopting the RI proposed by the manufacturer. Our opinion is based on an analysis of the RI values provided by the manufacturer of the reagents used by us, which applies only to selected age groups, without information whether the division into sex was taken into account, and despite the declaration of the use of the CLSI procedure, the insufficient number of reference groups.

Statistical analysis

It has been reported that log transformation failed to achieve both linearity of TSH and an acceptable normal distribution unless TPO-Ab positive subjects who add a right skew are removed prior to the analysis.

Did the authors examine a possible influence by contamination of the euthyroid sample, for instance with auto-antibody positive subjects?

Clinical categorisation is essential when establishing a reference range, which does not conceptionally extend to heterogeneous populations with various thyroid pathologies.

Answer:

We would like to thank the reviewer for drawing attention to the important problem of selecting patients for the determination of reference intervals mentioned in two paragraphs above. These issues are important in the process of determining RI and critical for direct methods. We got acquainted in detail with the available publications which used different criteria for excluding patients on the basis of available information such as ICD, taking medications, TPO antibodies. However, when planning our research, we were aware of the limitations of LIS functionality in our laboratory. It was not possible to download from the database results for single patient, which would allow obtaining a thyroid hormone profile for individual patients. Moreover, we also did not have access to the ICD and the identification of patients who had the same test repeatedly performed. This resulted in a lack of patient selection prior to statistical analysis. Therefore, we cannot provide clinical exclusion criteria for this study. Our analysis was based on the removal of outliners and the assumptions of Hoffman's method, according to which the linear part of the Q-Q plot corresponds to the range of empirical data whose distribution is consistent with the normal distribution. The Hoffmann method assume a Gaussian distribution of physiological test results and only this range of results is used to determine the reference intervals.

In order to prevent the doubts presented by the reviewers regarding the selection of the results analyzed in the Data Gathering chapter, we added information about the lack of any selection of the analyzed results. Information on the reason for this situation is included in the limitation at the end of the discussion.

The difficulties we have, encountered result from the need to import re-anonymized data. Therefore, attempts have already been made to adapt Laboratory and Hospital Information System to the possibility of importing the results assigned to an individual patient into the database, not only the results of a given parameter, without being related to other determinations performed on the same blood sample.

The appropriateness of the age groups should be evaluated with age as a continuous outcome to further confirm their physiologically-based selection by cut-off analysis. Given the large sample size this should be feasible.

Answer:

We did not decide to develop the data as a continuous variable, e.g. at intervals of 1 year, because it resulted in small a number of individual groups, especially in the youngest and the oldest age groups. That is why we decided to distinguish groups that correspond to the various stages of somatic and social development of a human being, i.e. infancy, preschool, school, adolescence and adult period, with a focus on the periods of young and old adulthood and senior age. The distinction of the main age ranges was also given for the assumed purpose of the study, i.e. the comparison of the RI values with the data from the reagent manufacturer, which are included in Tables 1 and 2. We developed the rationale for selecting just such age groups by adding an excerpt: “and reflects the main age groups for which the manufacturer provides reference value” in the Statistical Analysis chapter.

Tuckey test? Pls specify at what levels patients were excluded.

Answer: 

We used standard cut-off 1.5 IQR for detection and exclusion outliners. This information was added in the Statistical Analysis section.

The Hoffman method needs to be explained more in detail and referenced.

Answer:

As suggested by the reviewer, we have made significant changes to the Statistical analysis section and we hope that this will result in better readability of this chapter.

Why Hoffman? What is the difference to the method of Katayev el al.. The latter provides a detailed statistical procedure for dealing with the error term.

Answer:

The original Hoffman method involved a scatter plot of experiment values vs cumulative frequency in a probability scale, not a linear scale (cumulative frequency). As described by Dan Holmes et al. in 2018 (DOI: 10.1093/AJCP/AQY149), many authors presented a typical example of incorrect implementation of Hoffmann method using a linear cumulative frequency plot instead of probability scale like in Quantile-Quantile plots. 

We chose the Hoffman method having knowledge of the possibility of using the cumulative linear plot proposed by Katayev et al. (Katayev A, Balciza C, Seccombe DW. Establishing reference intervals for clinical laboratory test results: is there a better way?Am J Clin Pathol. 2010;133:180-186).

We also followed the discussion between Katayev and other authors in medical journals meticulously:

1.Katayev A, Fleming JK, Luo D, et al. Reference intervals data mining: no longer a probability paper method. Am J Clin Pathol. 2015;143:134-142 

2. Graham Jones, MD, Gary Horowitz, MD, Reference Intervals Data Mining: Getting the Right Paper, American Journal of Clinical Pathology, Volume 144, Issue 3, 1 September 2015, Pages 526–527, https://doi.org/10.1309/AJCP26VYYHIIZLBK

3. Holmes DT, Buhr KA. Widespread incorrect implementation of the Hoffmann method, the correct approach, and modern alternatives. Am J Clin Pathol. 2019;151:328-336

4. Alexander Katayev, MD, James K Fleming, PhD, Daniel T Holmes, MD, Kevin A Buhr, PhD, Widespread Implementation of the Hoffmann Method: A Second Opinion, American Journal of Clinical Pathology, Volume 152, Issue 1, July 2019, Pages 116–117, https://doi.org/10.1093/ajcp/aqz015

Due to the intended purpose of the work, which was, inter alia, to demonstrate that the RI determination by the Hoffman method can be achieved with the use of the simplest statistical packages (e.g. Excel, Statistica), we decided to use the original Hoffman method based on the use of quantile-quantile plots, preceded by removing outliers and visually and statistical fitting the regression line to empirical data and a hypothetical distribution. (For Excell package see Georg Hoffmann et al. in).

In the analyses carried out using the Hoffman method, we also obtained the error term in the regression equation. In our study, the curve fit was made on the basis of visual inspection and accepted on the basis of a linear correlation coefficient that could not be less than 0.99. Due to the almost perfect fit of our curves (r> 0.99), the component e of the regression equation was so insignificant that we did not report it in the publication. (Katayev and Larisch also did not provide the value of error)

FT4 does follow a normal distribution. Hence, a log transformation seems statistically unnecessary, and clinically unwanted, even potentially detrimental given the well known issues with interpretation of back transformed estimates in clinical medicine.

Answer:

As we mention above, we used log transformations in the study due to the log-normal distribution of fT4 values in all analyzed age groups. Perhaps this is due to the lack of strict exclusion criteria for people with thyroid disorders or taking medications that affect the regulation of thyroid secretion. We assume that these values were removed from the group of reference results in the course of further analysis. There are publications available confirming the reviewer's statement, but also those that indicate the log-normal distribution of fT4 results in both more and less numerous study groups. However, our results are consistent with reports by other authors, (see below) which indicates that both situations are possible depending on the studied population.

- Wang Y, Zhang YX, Zhou YL, Xia J. Establishment of reference intervals for serum thyroid-stimulating hormone, free and total thyroxine, and free and total triiodothyronine for the Beckman Coulter DxI-800 analyzers by indirect method using data obtained from Chinese population in Zhejiang Province, China. J Clin Lab Anal. 2017 Jul;31(4):e22069. doi: 10.1002/jcla.22069. Epub 2016 Sep 26. PMID: 27716997; PMCID: PMC6817203.

- Ganslmeier, Mira, Castrop, Claudia, Scheidhauer, Klemens, Rondak, Ina-Christine and Luppa, Peter B.. "Regional adjustment of thyroid hormone reference intervals" LaboratoriumsMedizin, vol. 38, no. 5, 2014, pp. 281-287. https://doi.org/10.1515/labmed-2014-0008

- Milinković N, Ignjatović S, Zarković M, Radosavljević B, Majkić-Singh N. Indirect estimation of reference intervals for thyroid parameters. Clin Lab. 2014;60(7):1083-9. doi: 10.7754/clin.lab.2013.130733. PMID: 25134375.

Figure 1 should be moved to the Results section. The extrapolation to the boundaries should be precisely indicated in the Figure, as this is a critical part of the method.

A statistical measure of how close the data are to the fitted line such as r squared should be reported.

Answer:

We moved the graph to the results chapter and modified it as suggested by the reviewer. We have introduced a regression equation describing the linear part of the graph with a correlation coefficient. We also added information to the statistical analysis chapter on the selection criteria for the linear part of the chart:

“The regression line included the middle range of data, which was initially fitted by visual inspection and was statistically confirmed by determining the linear correlation coefficient, only r> 0.99 was acceptable.”

Results

Line 192 A brief summary of main outcomes beyond merely referring to tables would be more informative to the reader, e.g. comment briefly on age dependency.

Answer:

A preliminary discussion of the reference interval distribution for TSH and fT4 is given in Figure 1. Table 1 is a retail numerical representation of the Figure 1 data for TSH along with a comparison of the percentage differences for each group with respect to the estimated RCV. The RCV values for the analyzed parameters are provided in the footer of the table.

We discuss the results of Table 1 in the text below for differences by age and gender. In order to improve the clarity of the discussion of the results of Table 1, we changed the order, starting with the discussion of age groups and then moving on to a comparison between sexes within the same age group.

Line 196 "decreased" with age" This could be more informative, for instance say something about the magnitude or relevance of effects. In a large sample there will always be some minor differences, which might not be all that relevant.

Line 200 "significant" Does this mean statistically significant?

Have differences been statistically assessed, for instance by interaction with age or age categories?

Answer:

All comparisons in our work were carried out in relation to the determined RCV value. Therefore, significant values are greater than the estimated RCV. We have quoted the RCV values calculated for our study at the end of the statistical analysis chapter

Discussion

The authors briefly discuss the method by Katayev et al.. This method has been successfully adopted by a few authors in the literature to verify TSH reference values, compare them with the limits provided by manufacturers and, importantly, facilitate clinical decision making in borderline conditions. The authors might want to consider discussing/ replicating some relevant findings from those publications, some of which they did not mention.

Answer:

As discussed above, we decided to use Q-Q charts in our work because of its application possibilities. at the same time, an ongoing discussion on the correctness of the Katayev method would expose us to critical comments from potential reviewers. Accordingly, the selected publication by Katayev et al. is cited to the same extent as works by other authors.

The discussion is detailed, but difficult to follow. For instance in line 346 the authors mention Ris and then a comparison with LRI.

A clearer structure and concise summary of similarities/ differences to other approaches would help.

Answer: 

Thank you very much for your comments on the difficulties with the structure and clarity of the discussion. Following the reviewer's guidelines, we have completely reorganized this chapter:

 - we limited our discussion only to the works of other authors based on indirect methods of RIs determination 

- the most important results from the available publications are summarized in table 5 - we removed too detailed quotations of other authors' results 

- the changed structure of the discussion consists of 3 parts: citing the most important results of our study, discussion of the results determined by the Hoffman method with the values provided by the reagent manufacturer, and finally, comparison of our results with data from international literature 

- we have identified the strengths and weaknesses of the study as separate subsections

We hope that such a structured chapter Discussion will meet the expectations of the reviewer and potential readers. We have made every effort to present our achievements in an unequivocal and legible manner.

Line 403 "therefore a large portion of the result comes from people without thyroid disorders." This raises some questions to be both examined (see Methods) and discussed. What impact do thyroid disorders have on the validity of the method? At what proportion of pathologies might the method fail? What makes the method robust?

Answer:

In response to a pertinent comment by the reviewer, we cited Drees et al. in a more precise way, and actually this sentence is:

“The legitimacy of determining reference intervals for thyroid hormones with an indirect method based on the results of hospital tests is additionally supported by the fact that their performance is very often ordered as a screening tests, and therefore a large portion of the results comes from people without thyroid disorders.”

Figures

I can’t seem to find legends to Figures.

Answer:

Descriptions of the figures were placed in the text in accordance with the guidelines for the authors, but they were barely visible due to the lack of separation from the text. In the revised manuscript, we placed the descriptions of the figures as separate paragraphs so that they are clearly visible and placed its one again at the end of the manuscript.

Reviewer #2: 

We kindly thank you for giving us the chance to improve our manuscript. Thank you very much for all your constructive comments concerning our manuscript. A detailed description of changes in the manuscript appears below.

The authors have defined reference intervals (RIs) for TSH and FT4 by applying the indirect Hoffman method based on a large data pools from patients and outpatients stored in laboratory information systems over a period of five years. There are not many data regarding the Hoffman method and thus, they can be compared only with manufacturers’ data. It is an analytical study. The methods should be reviewed by a statistician who has experience with Hoffman’s method.

The study is interesting and methodologically novel. However, I have some minor comments:

1. Line 400: “The legitimacy of determining reference intervals for thyroid hormones with

an indirect method…based on the results of hospital tests is additionally supported by

the fact that these tests are screened, and therefore a large portion of the results comes from people without thyroid disorders”. However, this method does not eliminate differences in the levels of parameters (TSH & FT4) resulting from drugs and concomitant diseases which are common in hospital sampling and interfere with TSH and particularly with FT4 measurement?

Answer:

We would like to thank the reviewer for drawing attention to the important problem of selecting patients for the determination of reference intervals. This is critically important for direct RI selection methods. We got acquainted in detail with the available publications which used different criteria for excluding patients on the basis of available information such as ICD, taking medications, TPO antibodies. However, when planning our research, we were aware of the limitations of LIS functionality in our laboratory. it was not possible to download from the single patient database, which would allow obtaining a thyroid hormone profile for individual patients. Moreover, we also did not have access to the ICD and the identification of patients who had the same test repeatedly performed. This resulted in a lack of patient selection prior to statistical analysis. For this reason, we cannot provide clinical exclusion criteria for this study. Our analysis was based on the removal of outliners and the assumptions of Hoffman's method, according to which the linear part of the Q-Q plot corresponds to the range of empirical data whose distribution is consistent with the normal distribution. The Hoffmann method assume a Gaussian distribution of physiological test results and only this range of results is used to determine the reference intervals.

In order to prevent the doubts presented by the reviewers regarding the selection of the results analyzed in the Data Gathering chapter, we added information about the lack of any selection of the analyzed results. Information on the reason for this is included in the limitation at the end of the discussion.

2. The data should be validated by other studies and compared with direct methods.

The URIs and LRIs estimated by the indirect method may be compared for concordance with the 2.5th and 97.5th percentile analysis? This evaluation can be performed easily and cheaply as validation of Hoffman’s method.

Answer:

Thank you for paying attention to this aspect of research and suggestions in the field of validation of the obtained results against other methods. We are aware that the validation of the results obtained by us would increase the credibility and confirm the application of the established reference intervals. However, due to the limitations in access to patients data presented in the previous answer, we did not have the possibility to create a group of patients homogeneously enough to be able to apply direct a posteriori methodologies based on the designation of the 2.5th and the 97.5th.

The difficulties we have encountered result from the need to import re-anonymized data. Therefore, attempts have already been made to adapt LIS and HIS to the possibility of importing the results assigned to a specific patient into the database, and not only the results of a given parameter, without being related to other determinations performed on the same blood sample.

3. The indirect method however, may not distinguish well enough pathological from non-

pathological levels and this may be an additional limitation.

Answer:

We are aware that no method of determining RI is perfect and it is always a choice between a well-defined but not very numerous group in direct methods and a less characterized but definitely more numerous group in indirect methods. In our opinion, any effort made to best match the reference interval to the studied patient population is better than adopting the RI proposed by the manufacturer. Our opinion is based on an analysis of the RI values provided by the manufacturer of the reagents used by us, which applies only to selected age groups, without information whether the division into sex was taken into account, and despite the declaration of the use of the CLSI procedure, the insufficient number of reference groups.

4. This analysis clearly shows the importance of measuring FT4 together with TSH in

establishing a diagnosis of thyroid disease. TSH is a reliable indicator of disease.

Answer:

Obviously, we agree with the opinion of the reviewer, which is consistent with the current recommendations cited at the end of Background section of our manuscript:

“According to the current recommendations, in order to screening for thyroid primary dysfunction, first TSH determinations should be performed repetitively in 3-6 month intervals, followed by fT4 for differentiation of subclinical and “overt” thyroid dysfunction. fT3 determinations should be ordered only in specific cases.”

5. A minor linguistic revision is needed.

Answer:

We have made the necessary changes to the text of the manuscript.

---

## [Editor Report · Decision Letter 1]

9 Dec 2021

Establishing laboratory-specific reference intervals for TSH and fT4 by use of the indirect Hoffman method

PONE-D-21-26253R1

Dear Dr. Płaczkowska,

We’re pleased to inform you that your manuscript has been judged scientifically suitable for publication and will be formally accepted for publication once it meets all outstanding technical requirements.

Kind regards,

Joseph DiStefano III, PhD

Academic Editor

PLOS ONE
---

## [Editor Report · Acceptance letter]

29 Dec 2021

PONE-D-21-26253R1 

Establishing laboratory-specific reference intervals for TSH and fT4 by use of the indirect Hoffman method 

Dear Dr. Płaczkowska:

I'm pleased to inform you that your manuscript has been deemed suitable for publication in PLOS ONE. Congratulations! Your manuscript is now with our production department. 

Kind regards, 

on behalf of

Prof. Joseph DiStefano III 

Academic Editor

PLOS ONE